# Assessing the Presence of a Monoculture: From Definition to Quantification

Silvio Franco [1], Barbara Pancino [1,*], Angelo Martella [1] and Tommaso De Gregorio [2]

[1] Department of Economics, Engineering, Society and Business Organizations, University of Tuscia, Via del Paradiso 47, 01100 Viterbo, Italy

[2] Ferrero HCo Hazelnut Company, Korvella-Ferrero, Str. S. Valentino, 01032 Caprarola, Italy

* Correspondence: bpancino@unitus.it

**Abstract:** The term monoculture is widely used in the scientific literature concerning the agricultural sector. However, it is very difficult to find a clear and shared definition of this term. This study investigates the concept of monoculture in agricultural areas where high specialization in a specific crop is observed. Therefore, we refer to a *territorial-level* definition, which associates the idea of monoculture to the prevalent presence of a crop in a region including many farms. The objectives of the paper are: (i) to define indicators capable of verifying the existence of this condition; (ii) to test the ability of such indicators in identifying the effective presence of a monoculture. A set of Italian areas identified as monoculture in the recent literature were selected to carry out a quantitative analysis, assessing different indexes of monoculture. On the basis of the obtained results, such an analysis should help in comparing the monoculture indexes and fostering a discussion on their suitability and descriptive capacities.

**Keywords:** monoculture index; crop concentration; crop diversification; Shannon index; Italian monoculture





## 1. Introduction—Definition of Monoculture

The term monoculture is widely used in the scientific literature concerning the agricultural sector. A search performed in the Scopus database (www.scopus.com, accessed on 20 May 2021) reports 1388 papers with this word in the title. Even if the concept of monoculture is also used in social sciences, the largest proportion of such papers concerns the subject "agricultural and biological sciences" (949 papers) and/or "environmental sciences" (412). However, despite this widespread popularity, it is very difficult to find a clear and shared definition of this term. This is probably because the idea of monoculture appears quite intuitive, as it might seem redundant to define a term whose meaning is considered obvious. However, if we examine this concept carefully, we realize that several aspects are worth investigating further. Indeed, an unclear definition of what a monoculture effectively is has significant implications not only from a scientific perspective but also in public communication. This is true, in particular, for the negative meaning attributed to the term monoculture in relation to the possible environmental impacts of agriculture, first of all the risk of biodiversity loss. On the other hand, productive specialization, as the monoculture prefigures, is associated with the economic advantages originated by returns to scale, economies of agglomeration, concentrations of supply and the strengthening of local supply chains.

The first concept to address is whether monoculture refers either to space or time. In other terms, it should be clarified whether a monoculture condition is due to a replication of the same crop in the same field for a certain number of years or to the exclusive, or largely dominant, presence of a specific crop in a wide agricultural area.

To this concern, the definitions of monoculture offered by different dictionaries are quite general and not useful enough. Luckily, the explanation of the concept provided by [1], who define crop monoculture "*as the practice of replanting the same crop species in the*

*same field, with no break to a different crop*", is surely more helpful. However, the same authors continue: "*crop monoculture is also used to describe large areas planted to the same crop species*". Then, they conclude "*we mean crop monoculture in the temporal and not the spatial sense*". So, according with this definition, we could argue that a monoculture can exist either in space or in time. Many other authors have a similar perspective (see, for example, [2–4]), which focus their attention at a field or a farm level and, as an implicit consequence, limit the monoculture condition to the time dimension.

This is not the opinion of [5], who state: "*cultivation of a single crop over a large area for consecutive years is the agronomic definition of monoculture*". The idea that a monoculture condition is characterized by an intrinsic spatial connotation, which goes beyond the single-field/farm dimension, has at least two important consequences. The first one is that the concept can be extended to multi-year crops, such as all tree crops, which are "monocultures in time" by definition. The second one is related to the fact that a monocultural condition can be reduced, improving diversity both in time, introducing crop rotations, and in space, inserting alternative crops or inter/mixed cropping.

To summarize, there are two different interpretations of the term: a *farm-level* definition, which identifies monoculture as a repeated cultivation of the same plant species for several years on the same land, and a *territorial-level* definition, which associates the idea of monoculture to the prevalent presence of a crop in a region including a large number of farms. This latter interpretation, therefore, identifies a high level of specialization on a single crop production within an agricultural region.

The first interpretation is more related to agronomic farm management, while the second one has significant implications in economic and environmental terms in a wider perspective. For this reason, in this study, we will refer exclusively to the latter meaning of the term monoculture, with a twofold objective: (i) to define indicators capable of verifying the existence of such a condition within an agricultural area; (ii) to test the ability of such indicators in identifying the effective presence of a monoculture. Concerning the latter objective, the study was carried out by evaluating some highly specialized Italian agricultural areas; this choice is justified, apart from the authors' direct experiences and knowledge, by the fact that such areas are identified as monoculture in recent scientific and newspaper articles (see Section 4).

## 2. Monoculture Indexes

In the scientific literature, it is not easy to find specific studies describing methods or tools able to assess the presence of a monoculture condition within a given territory. The main reason could be that monoculture is generally considered as a "basic" concept, which does not deserve a specific definition and measurement. This point, as briefly discussed in the introduction, it is not so trivial, especially when the concept of monoculture refers to a spatial dimension.

Indeed, apart from exceptional cases, a situation where a single crop covers all the agricultural land of a region never occurs. It is not uncommon, however, that a crop, occupying a significant portion of the agricultural area, assumes a preeminent role compared to other ones. For this reason, it does not seem scientifically acceptable to distinguish, simply on the basis of a personal feeling, if a specific area falls under the definition of monoculture.

It appears more suitable to produce an objective measure of the level of monoculture and, when required, to establish a specific threshold identifying the existence of such a condition.

To assess a level of monoculture, it is useful to refer to the concepts of (productive) concentration and diversification. Indeed, it is quite evident that a higher presence of monoculture is associated with a greater level of crop concentration and a lower level of crop diversification. Several indexes can be used to evaluate crop concentration/diversification, the value of which is based on: (1) the number of crops in the region and (2) the relative dimension of each one of these crops with respect to the total agricultural land in the region itself.

When $N$ is the number of crops present in a territory and $p_i$ ($I = 1 \ldots N$) is the share of agricultural area occupied by the generic cultivation i, with respect to the total agricultural area ($\Sigma p_i = 1$), the following crop concentration/diversification indexes can be defined:

The following indexes have been considered and their definitions are reported in [6,7]:

1. Herfindahl–Hirschman concentration index:

$$I_H = \sum_{i=1\ldots N} (p_i)^2 \tag{1}$$

2. Ogive concentration index:

$$I_o = 1 - N \times \sum_{i=1\ldots N} \left[ p_i - \left( \frac{1}{N} \right) \right]^2 \tag{2}$$

3. Gini concentration index, whose calculation requires an ascending sorting of single-crop areas $y_i$ ($i = 1 \ldots N$). The Gini index assessment can be performed in different ways, which give the same results:

$$I_o = 1 - N \times \sum_{i=1\ldots N} \left[ p_i - \left( \frac{1}{N} \right) \right]^2 \tag{3}$$

$$I_{G1} = \frac{(N + 1 - 2 \times \sum_{i=1\ldots N} C_i)}{(N - 1)} \tag{4}$$

$$I_{G2} = \left[ \frac{1}{N-1} \right] \times \left[ N + 1 - 2 \times \left( \frac{\sum_{i=1\ldots N}(N + 1 - i) \times y_i}{\sum_{i=1\ldots N} y_i} \right) \right] \tag{5}$$

$$I_{G3} = \frac{\left( 1 - 2 \times \sum_{i=1\ldots N} \frac{[(c_i - c_{i-1}) \times (p_i - p_{i-1})]}{2} \right)}{\left( 1 - \frac{1}{N} \right)} \tag{6}$$

where $c_i$ is the cumulative relative frequency of $p_i$.

4. Shannon diversification index:

$$I_s = - \sum_{i=1\ldots N} [p_i \times \ln(p_i)] \tag{7}$$

This index, sometimes identified as the Entropy Index, has a value ranging from 0 (maximum concentration) to $\ln(N)$ (maximum diversification). To normalize it, two different formulas, which give the same result, can be adopted:

$$I_{Sn} = - \sum_{i=1\ldots N} [p_i \times \log_N(p_i)] \tag{8}$$

$$I_{Sn} = - \frac{(\sum_{i=1\ldots N} [p_i \times \ln(p_i)])}{\ln(N)} \tag{9}$$

To use this diversification index as a monoculture index, it is necessary to calculate its complement; it brings one to the following (modified) formulation of the Shannon index:

$$I_{Sc} = 1 + \frac{(\sum_{i=1\ldots N} [p_i \times \ln(p_i)])}{\ln(N)} \tag{10}$$

Among these indexes, the most used in the literature for the evaluation of crop concentration is the Shannon index, although in most cases it is applied in its original formulation of the diversification index. Indeed, a recent study [8] states that "*in the context of crop production [the Shannon index] measures the crop diversity by representing the number of crop types and evenness of the area covered by the crops*" and concludes that "*it is important,*

*from an ecological perspective, to increase the Shannon index for crop diversity*". Meanwhile, [9] state that "*metrics such as the Shannon index are suitable to quantify the compositional diversity of land-use options* [10], *where Shannon's value accounts for the number and proportions of land-use types*". Furthermore, the Shannon index was also used in some studies focusing on the economic implication of crop diversification [11–14].

More limited is the utilization of the Gini coefficient as an index of cropping diversity. An example is represented by the study of [15], where "*a Gini coefficient value of zero indicates complete diversity (or equal crop concentration), with equal area in each crop, and a value of one indicates complete concentration in one crop*".

All four mentioned indexes were applied and compared in their ability in describing crop diversification in [7] and, more recently, in [6].

## 3. The Constraints of Limiting the Area

A key aspect in identifying the condition of monoculture, and consequently in assessing its level of effective presence through the concentration/diversification indexes illustrated in the previous section, is represented by the region delimitation.

This delimitation can be difficult when the region under investigation is not clearly defined ex ante. Indeed, it is evident that when we consider a region characterized by a high specialization, the level of crop concentration tends to progressively reduce, moving away from the area where the crop itself shows the maximum density. In other words, a regional delimitation that represents an efficient trade-off between the need to include the most significant presence of the crop and the ability to circumscribe where the crop is "prevalent enough" compared to all the others should be chosen.

Indeed, focusing on the area where the cultivation shows a very high presence, the monoculture (crop concentration) indexes will assume greater value, but, at the same time, the limited extension of the region will determine less significant economic and environmental implications. On the contrary, if we expand the border of the region where key production takes place, it is inevitable that the indexes of production specialization tend to progressively reduce, weakening the territorial effects of the monoculture.

To better explain this concept, for example, we can consider a generic crop with a total dimension of 12,500 ha inside a hypothetical region. Within this region, we can define three possible areas of increasing size (small, medium and large) and, consequently, progressively smaller production specialization (Table 1).

**Table 1.** Example of the effects on level of monoculture of different regional dimension.

| Region | Area (ha) | Agricultural Area (ha) | Crop Area (ha) | Crop Area/ Agricultural Area | Crop Area/ Total Crop | Shannon Index |
|---|---|---|---|---|---|---|
| Small | 10,000 | 8000 | 6000 | 75% | 48% | 0.60 |
| Medium | 20,000 | 15,000 | 9000 | 60% | 72% | 0.45 |
| Large | 40,000 | 30,000 | 12,000 | 40% | 96% | 0.25 |

As it can be observed, when the size of the area is limited to the most intensely planted area (where the share of the agricultural area reaches 75%), the concentration index assumes a high value (0.60), which means that it recognizes an actual situation of monoculture. However, only a limited share of the entire crop area falls into this region (48%) and, consequently, it does not appear correct to identify it as the local production area. On the contrary, if we consider the largest area, where almost all (96%) of the local crop area falls, the share of the agricultural area destined for cultivation decreases (40%), and the concentration index drops to a value (0.25) which cannot be associated with a monoculture condition. The medium area identifies a situation in which the crop takes on a preponderant role compared to other crops (60%) and, at the same time, represents a significant share of the total local production (72%).

It follows that the definition of the area represents a key issue in order to obtain significant results through the most common concentration indexes and, consequently, allow the identification of an effective monoculture situation.

## 4. Examples of Possible Monocultures in Italy

Italy, despite its high product diversification in agriculture, shows several situations of crop specialization determined by several reasons, such as climatic characteristics, historical preferences and economic matters. In this situation, it may be interesting to calculate crops' concentration indexes and compare the level of monoculture in different areas of the country.

Among the crops showing a tendency to concentrate in particular areas of the country, the choice of areas in which carry out a quantitative analysis was based on bibliographic research in which the most widespread cultivation was somehow defined as "monoculture". Indeed, it is worth mentioning the following: vine [16–18], apple [19,20], hazelnut [21–24], durum wheat [25,26] and rice [27,28].

Considering these crops, the assessment of monoculture indexes was carried out in the following regions (see Figure 1):

Vine (Prosecco)—region: Veneto (Province: Treviso);
Vine (Soave)—region: Veneto (Province: Verona);
Vine (Chianti)—region: Tuscany (Provinces: Florence–Siena);
Vine (Montepulciano)—region: Abruzzo (Province: Chieti);
Apple—region: Trentino Alto-Adige (Province: Trento);
Hazelnut—region: Piedmont (Province: Cuneo);
Hazelnut—region: Lazio (Province: Viterbo);
Hazelnut—region: Campania (Provinces: Avellino–Napoli);
Durum wheat—region: Puglia (Province: Foggia);
Rice—region: Piedmont (Provinces: Vercelli–Novara).

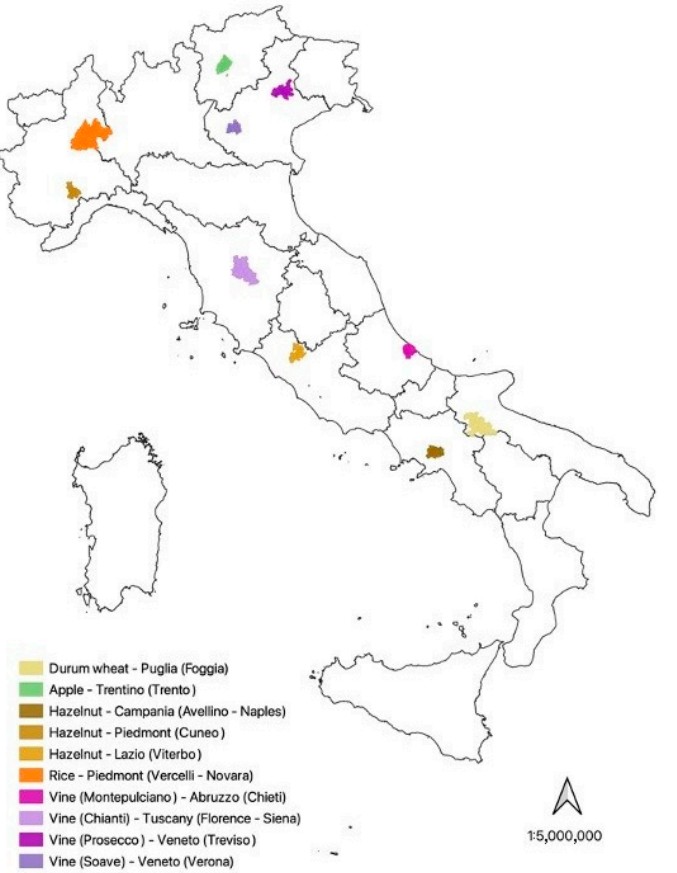

**Figure 1.** Regions selected as possible monocultures.

The first column of Table 2 reports the total area of the region where a single crop can be considered as a monoculture. Each one of such regions was bordered after a careful evaluation based on the issues that arose in the previous section. The last column of Table 2

shows the share of the crop area with respect to the whole agricultural area of each selected region; this figure ranges from 37% to 84%, pointing out a high presence of the crop within the selected region. Table 3 shows the values of the four concentration indexes in the selected regions.

**Table 2.** Total, agricultural and selected crop area in each region.

| Region | Total Area (ha) | Agricultural Area (ha) | Crop Area (ha) | Crop Area/ Agricultural |
|---|---|---|---|---|
| Vine—Prosecco (Veneto) | 45,903 | 16,921 | 8131 | 48.1% |
| Vine—Soave (Veneto) | 30,327 | 18,635 | 12,326 | 66.1% |
| Vine—Chianti (Tuscany) | 95,702 | 31,909 | 11,778 | 36.9% |
| Vine—Montepulciano (Abruzzo) | 28,377 | 17,649 | 12,340 | 69.9% |
| Apple (Trentino) | 37,351 | 10,067 | 6320 | 62.8% |
| Hazelnut (Piedmont) | 29,458 | 11,197 | 4741 | 42.3% |
| Hazelnut (Lazio) | 37,978 | 21,266 | 14,008 | 65.9% |
| Hazelnut (Campania) | 35,244 | 11,480 | 7546 | 65.7% |
| Durum wheat (Puglia) | 93,014 | 68,483 | 50,874 | 74.3% |
| Rice (Piedmont) | 160,813 | 124,226 | 104,585 | 84.2% |

Source: Italian Agricultural Census, 2010 (ISTAT. Available online: https://www.istat.it/it/censimenti-permanenti/censimenti-precedenti/agricoltura/agricoltura-2010, accessed on 15 April 2021). (www.istat.it).

**Table 3.** Monoculture (concentration) indexes in selected regions.

| | Shannon Index | Gini Index | Herfindahl Index | Ogive Index |
|---|---|---|---|---|
| Vine—Prosecco (Veneto) | 0.394 | 0.780 | 0.300 | 0.247 |
| Vine—Soave (Veneto) | 0.497 | 0.809 | 0.452 | 0.420 |
| Vine—Chianti (Tuscany) | 0.346 | 0.758 | 0.227 | 0.176 |
| Vine—Montepulciano (Abruzzo) | 0.595 | 0.878 | 0.521 | 0.487 |
| Apple (Trentino) | 0.563 | 0.833 | 0.514 | 0.417 |
| Hazelnut (Piedmont) | 0.267 | 0.647 | 0.233 | 0.174 |
| Hazelnut (Lazio) | 0.454 | 0.778 | 0.453 | 0.399 |
| Hazelnut (Campania) | 0.488 | 0.820 | 0.457 | 0.407 |
| Durum wheat (Puglia) | 0.600 | 0.866 | 0.560 | 0.537 |
| Rice (Piedmont) | 0.747 | 0.941 | 0.717 | 0.698 |
| Mean | 0.495 | 0.811 | 0.443 | 0.396 |
| Standard deviation | 0.140 | 0.079 | 0.153 | 0.163 |
| Variation coefficient | 28.2% | 9.8% | 34.6% | 41.3% |

Source: our elaboration on Italian Agricultural Census data (2010).

A first aspect to be considered is the high level of accordance within the different indexes, which, besides minor diversities, rank the areas within a similar range, despite the differences in crop or location. This trend, as shown in Table 4, is confirmed by the correlation coefficients between the indexes that are always greater than 0.90. A different situation can be observed for their absolute values, which show dissimilar means and variation coefficients.

**Table 4.** Correlation matrix.

| | Shannon | GINI | Herfindal | Ogive |
|---|---|---|---|---|
| Shannon | 1.000 | | | |
| GINI | 0.967 | 1.000 | | |
| Herfindahl | 0.976 | 0.903 | 1.000 | |
| Ogive | 0.973 | 0.909 | 0.992 | 1.000 |

The Gini index, beyond assuming higher values, is characterized by limited variability. It follows that this index, overestimating the crop concentration and being unable to sufficiently discriminate situations that are quite dissimilar, is not an ideal indicator to evaluate the presence of a monoculture condition.

An opposite situation can be observed for the Herfindahl and Ogive indexes, which convey very similar information (their correlation coefficient is greater than 0.99), even if the latter has lower and more scattered values. Both these indexes show some limits in assessing the monoculture level, because their values are too different even when they are used to evaluate similar situations.

The Shannon index, in the formula proposed in this study, appears to be the most suitable to assess the monoculture level within a region. The discriminating capacity among the situations proposed in this case study is stronger than the other indexes considered. Nevertheless, the correlation between the crop area/agricultural area and the result of the Shannon index is high (0.92). The difference between the two measures is related to the number and size of secondary crops; in addition, this high value is explained by the fact that all regions chosen for the analysis where, ex ante, they are candidates as monocultural areas. Furthermore, the fact that its average value is about 0.5 suggests, at least in general terms, an idea regarding the possible threshold to discriminate actual monocultures from other situations which, even if characterized by a significant productive specialization, do not reach a critical level of crop concentration. Such properties can be observed in Figure 2, where the Shannon index values for the selected regions are graphically compared.

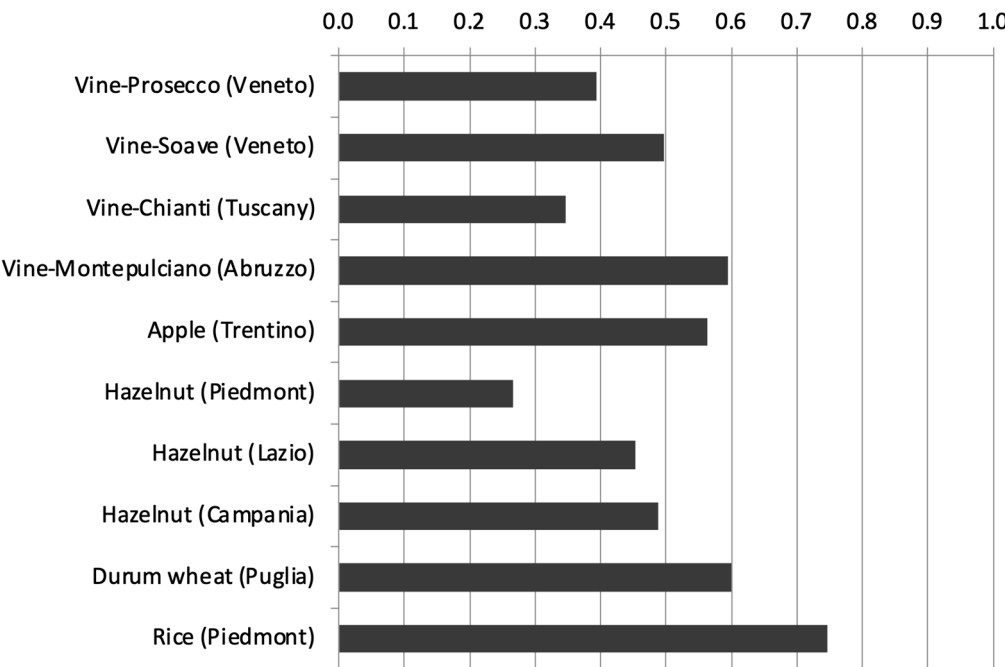

**Figure 2.** Values of monoculture (Shannon) index in the selected region.

An important aspect to mention concerns the static nature of the monoculture level assessed through these indexes; indeed, they are based on the current crops' dimension in the analyzed region, without contemplating the changes due to the expansion or narrowing of the main crop surface.

This point, which must be carefully considered for annual crops, is particularly critical in highly specialized regions where the expansion of the main crop has recently varied due to different factors, mainly the product market.

To support this consideration, in Table 5, the variation between 2010 and 2020 of the main crops in the selected regions is reported (the 2020 data are available from the Italian census by a sample survey and only at provincial level). It can be observed how, in some areas, the main crops reduced their presence, while others did not show appreciable changes. It is evident how four regions reported a significant increase: here, the crop concentration value certainly increased, substantially modifying the local monoculture situation.

**Table 5.** Area variation in the main crops in the provinces of the selected regions.

| Monoculture | Crop Area 2010–2020 |
| --- | --- |
| Vine—Prosecco (Veneto) | 36.2% |
| Vine—Soave (Veneto) | 0.9% |
| Vine—Chianti (Tuscany) | −2.2% |
| Vine—Montepulciano (Abruzzo) | −1.0% |
| Apple (Trentino) | −5.4% |
| Hazelnut (Piedmont) | 41.1% |
| Hazelnut (Lazio) | 27.2% |
| Hazelnut (Campania) | 24.4% |
| Durum wheat (Puglia) | 3.9% |
| Rice (Piedmont) | −7.8% |

Source: our elaboration on Italian Agricultural Census data (2010, 2020).

The results strengthen what was stated in Section 2, where it was emphasized how the Shannon index has been used by most scholars assessing crop concentration/diversification. Furthermore, the suitability of the Shannon index as an indicator of crop monoculture is also confirmed by the comparison carried out in recent reviews [6,7].

On the basis of the figures reported in Figure 2, the implication of the choice of a monoculture threshold equal to 0.5 can be analyzed. In particular, this value seems to highlight the situations of higher crop concentration (rice in Piedmont, durum wheat in Puglia, vine in Abruzzo and apple in Trentino) from the other ones. However, it should be noted that such a rigid discrimination excludes some crops from a possible monoculture condition (vine in Veneto and hazelnut in Lazio and Campania) which show a considerable concentration with Shannon index values between 0.4 and 0.5.

The consequence of a predefined threshold, such as 0.5, must be carefully evaluated in relation to the static nature of the monoculture level assessment. Indeed, in our case, some crops with a a Shannon index lower than 0.5 and showing a significant increase in their dimension in 2010 (vine—prosecco and hazelnut in all regions) could have overcome the threshold value, assuming the condition of monoculture today.

The reflections suggested in this work, which represents a first attempt to strengthen the scientific connotation of the concept of monoculture, have some limitations that may be overcome in future research on the subject. Among these, a considerable one is the exclusive reference to the agricultural dimension without taking into account the incidence of the agricultural area with respect to the total ecological space. Indeed, it is evident that the idea of monoculture and its implications in relation to the economic and environmental dimensions cannot ignore the weight (in spatial terms) of agriculture with respect to other land uses, in particular those providing ecosystem services, such as forests and natural and semi-natural spaces.

## 5. Conclusions

In this study, we tried to gain insight into the concept of monoculture, a term that is commonly used to identify agricultural areas where high crop specialization is observed. The discussion focused on three key aspects linked to the term monoculture: (i) a clear definition; (ii) the delimitation of the region; (iii) indicators able to evaluate the effective presence of a monoculture in a specific region. In particular, as far as it concerns the last issue, a quantitative analysis was carried out on a set of Italian areas identified as monocultures in the recent literature. Considering these areas, different indexes were assessed, comparing their suitability and descriptive ability on the basis of the obtained results.

The various concerns mentioned in the study tried to clarify some crucial points, with the aim to apply the term "monoculture" with more thoughtfulness and awareness. We think that such goal deserves to be pursued, and a higher scientific basis to a term so often used, and sometimes abused, in the literature should be provided. The presence of monocultures is often associated with specific economic implications and environmental impacts, as mentioned in the introduction section; this can lead to scientific misunderstandings

among scholars and, in addition, dangerous misinformation in the public opinion, which can result in different interpretations according to personal ideals.

**Author Contributions:** Conceptualization, S.F., B.P. and A.M.; methodology, S.F.; software, A.M.; validation, S.F., B.P., A.M. and T.D.G.; formal analysis, S.F. and A.M.; investigation, S.F. and B.P.; resources, S.F., B.P., A.M. and T.D.G.; data curation, A.M.; writing—original draft preparation, S.F.; writing—review and editing, B.P. and T.D.G.; visualization, A.M.; supervision, S.F.; project administration, B.P.; funding acquisition, S.F. and B.P. All authors have read and agreed to the published version of the manuscript.

**Funding:** This research was funded by Ferrero Trading Lux S.A. through the project Hazel-TIC Territorial Impact of Hazelnut Cultivation (2020–2021).

**Institutional Review Board Statement:** Not applicable.

**Informed Consent Statement:** Not applicable.

**Data Availability Statement:** Data available in a publicly accessible repository that does not issue DOIs. Publicly available datasets were analyzed in this study. This data can be found here: https://www.istat.it/it/censimenti-permanenti/censimenti-precedenti/agricoltura/agricoltura-2010.

**Conflicts of Interest:** The authors declare no conflict of interest.

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
