# Peer review of "Assessing the Presence of a Monoculture: From Definition to Quantification"

_agriculture, doi:10.3390/agriculture12091506_

Round 1

Reviewer 1 Report

This paper presents an interesting case regarding the definition of monoculture as it utilised in different disciplines. More importantly as authors mention, the term monoculture is ambiguous in spatial and temporal context. The authors provide good examples from the literature, however, it was perhaps more useful to insert other cropping systems in the analysis and discussion and compare and contrast their merit in the chosen areas (if they exist). Also it is not very clear whether or not management systems such as fallow is considered in the definitions utilised in the paper. Authors need to elaborate more on the indicators that they have used to choose the monoculture areas to test the indices and how can we related the crop area/agricultural area to the indices and the result of Shanon index. Apart from the above ratio what else do we know about the chosen areas (such as the number of other reported crops etc.) that makes them a candidate for a ‘concentrated’ monoculture?.

The discussion section should be more adequately expanded and relate to the work that others have done in this area. 

The article require some editing both in terms of structure (many short paragraphs, lack of coherence and  vague references such as ‘second paragraph’ and proper grammar and style. It is often difficult to find out what the authors intended to say. 

Some specific comments below:

Line 10: Abstract has only one paragraph

Line 24: please add the URL to the Scopus database or reference it properly in the bibliography section. 

Line 28-29: Why this sentence is made into a separate paragraph?

Line 26: concerns should not be plural

Line 54: what about other systems such as intercropping and mixed-cropping? 

Line 68: where is paragraph 4. 

Line 74-75: discussed

Lines 91-114: please use proper software to write down the formula

Line 99: Can be

Lines 137-143: This paragraph is not clear to me. Please re-write it. 

Line 175-6: what is the meaning of ‘productive vocation’?

Line 204: define or reference ISTAT properly

Figure 1: please make a proper map with scale, north arrow. The caption should be below the figures. 

Tables 1 and 2: please combine these tables or at least add the crop name and crop area/agriculture to table 2. 

Line 180: what was the criteria for choosing these regions? Do you consider these areas pure monoculture discuss a bit more? 

Line 213: please show this visually or in a table. 

Line 243: Is it possible to calculate the value of Shanon index for all these year? if yes please add that analysis. 

Reviewer 2 Report

This is an interesting paper which aims to point out the inconsistency with the definition of “monoculture” in agricultural setting. While the concept laid out in the paper is novel and interesting, some crucial information is missing. One of the major concerns is that the layout of the paper is not consistent such that the sections are not clear. It is hard to distinguish whether it is a review paper or a research paper. The quantitative analysis is done with limited data, but the methods are not clear.  

Abstract: The purpose of this study has not been defined clearly.  Line 16-19: Clear objective is not provided

Introduction: Line 31: Why is it worth investigating further? The introduction section about the “Definition of monoculture” misses to explain what kind of issue is being caused due to the inconsistency in the definition of “monoculture’. This makes the paper confusing.

Line 33-35: Please rephrase this as a question.

Line 61: The reasons discussed are not clear. Does this mean that the author chose one of the already given definitions for monoculture? How is this justified in the case where monoculture is also defined as planting a crop in a given field for multiple years? Is this not again leaving out the concept behind one of the definitions of “monoculture”? This contradicts the purpose of the paper itself.

Section 2. Defining monoculture indexes: Is this a part of the introduction? Most of the indexes are not defined.

Line 116: Why is Shannon index the most used index?

Line 118: Please specify the concern being mentioned?

Discussion

Line 251: Rather than stating a paragraph, it is better to put the exact information.

Conclusion

Line 290 – 292: What are the economic implications? This is an important aspect to justify the purpose of the study and should be highlighted throughout the study.

Round 2

Reviewer 1 Report

The manuscript is generally improved and the message is more clear now. 

However, I still believe that the article needs minor revision in terms of its English language. This will help with the readability of the article.  

The formulas should be presented in a proper way and the map in Figure 1 must follow the cartographic principals (if MDPI thinks that it is necessary). 
